# A polycistronic transgene design for combinatorial genetic perturbations from a single transcript in *Drosophila*

Alexander G. Teague [¤a], Maria Quintero, Fateme Karimi Dermani, Ross L. Cagan [¤b], Erdem Bangi *

Department of Biological Science, Florida State University, Tallahassee, Florida, United States of America

¤a Current address: Adicet Bio, Redwood City, California, United States of America
¤b Current address: Institute of Cancer Sciences, University of Glasgow, Glasgow, Scotland, United Kingdom
* ebangi@bio.fsu.edu

**Data Availability Statement:** All relevant data are within the manuscript and its Supporting Information files.

## Abstract

Experimental models that capture the genetic complexity of human disease and allow mechanistic explorations of the underlying cell, tissue, and organ interactions are crucial to furthering our understanding of disease biology. Such models require combinatorial manipulations of multiple genes, often in more than one tissue at once. The ability to perform complex genetic manipulations *in vivo* is a key strength of *Drosophila*, where many tools for sophisticated and orthogonal genetic perturbations exist. However, combining the large number of transgenes required to establish more representative disease models and conducting mechanistic studies in these already complex genetic backgrounds is challenging. Here we present a design that pushes the limits of *Drosophila* genetics by allowing targeted combinatorial ectopic expression and knockdown of multiple genes from a single inducible transgene. The polycistronic transcript encoded by this transgene includes a synthetic short hairpin cluster cloned within an intron placed at the 5' end of the transcript, followed by two protein-coding sequences separated by the T2A sequence that mediates ribosome skipping. This technology is particularly useful for modeling genetically complex diseases like cancer, which typically involve concurrent activation of multiple oncogenes and loss of multiple tumor suppressors. Furthermore, consolidating multiple genetic perturbations into a single transgene further streamlines the ability to perform combinatorial genetic manipulations and makes it readily adaptable to a broad palette of transgenic systems. This flexible design for combinatorial genetic perturbations will also be a valuable tool for functionally exploring multigenic gene signatures identified from omics studies of human disease and creating humanized *Drosophila* models to characterize disease-associated variants in human genes. It can also be adapted for studying biological processes underlying normal tissue homeostasis and development that require simultaneous manipulation of many genes.

**Funding:** This work was supported by grants R03 CA219321 (EB) and R21 GM141734 (EB) from the National Institutes of Health. The funders had no role in study design, data collection, and analysis, decision to publish, or preparation of the manuscript.

**Competing interests:** The authors have declared that no competing interests exist.

## Author summary

The rapid advancement and increased accessibility of genome profiling and big-data approaches have provided unprecedented access to human disease genome landscapes, opening the door to personalized approaches to disease diagnosis and therapy. These efforts continue to identify increasingly complex mutation profiles and gene expression signatures correlating with specific disease outcomes and responses to treatment, emphasizing the importance of experimental models that reflect the genetic complexity and diversity of human disease to investigate the functional relevance of these complex signatures and molecular mechanisms underlying their effects. Here we describe a flexible and streamlined strategy to introduce multiple genetic perturbations to cells in a cell-type and tissue-specific and temporally controlled manner from a single transgenic construct inserted into a predetermined genomic location in Drosophila. This new design represents a valuable addition to the arsenal of tools available in Drosophila to generate more representative human disease models, functionally interrogate individual human disease-associated gene variants and more complex profiles that include alterations in multiple genes.

## Introduction

Most common human diseases are polygenic; they arise due to the disruption of multiple genes and are also influenced by environmental factors [1]. For instance, genome landscapes of most solid tumors include concurrent somatic alterations in numerous oncogenes and tumor suppressors and many germline variants that can modify disease progression and response to therapy [2–4]. The ability to capture the genetic complexity and heterogeneity of human disease in experimental models is vital for the functional exploration of cancer omics datasets, improving our mechanistic understanding of disease biology and developing effective therapies. In addition to targeting multiple genes, *in vivo* disease modeling requires transgenes for tissue or cell-type specific and temporally regulated genetic manipulations and fluorescent labeling of targeted cells to study their interactions with their environment. Furthermore, parallel genetic perturbations within the local microenvironment or distant organs are often necessary to explore short-range and systemic interactions mediating disease phenotypes. While multiple such tools are available in genetically tractable model systems like *Drosophila* [5], bringing all these genetic tools together in experimental animals is challenging. It represents a significant barrier to *in vivo* disease modeling.

 The most commonly used approach for genetic manipulations in *Drosophila* is the targeted transgene expression platform UAS/Gal4/Gal80$^{ts}$[6,7]. In this method, genetic elements are cloned downstream of the Gal4 inducible UAS promoter, and transgene expression is induced using transgenic Gal4 lines with tissue-specific expression patterns. Temporal control of expression is achieved by Gal80$^{ts}$, a temperature-sensitive allele of Gal80 that inhibits Gal4 activity [7]. Thousands of transgenic *Drosophila* Gal4 lines with tissue and cell-type-specific expression patterns and genome-wide collections for ectopic expression, knockdown, and editing of most genes in the *Drosophila* genome are publicly available [8–11]. We have previously used these transgenic resources to establish multigenic models of colorectal cancer by genetically manipulating *Drosophila* orthologs of genes recurrently mutated in human colon tumors [12]. These models were created by combining multiple transgenic insertions into a single genetic background using standard *Drosophila* genetic crosses. The presence of a large number of transgenic constructs—each inserted in a different genomic location—makes

building and maintaining the stability of experimental animals a significant challenge and limits the additional genetic manipulations that can be performed for mechanistic studies.

To overcome these challenges, we modified the pUAST-attB vector, which allows targeted integration at predetermined locations within the *Drosophila* genome [13], to create a multigenic vector, pWALIUM 3xUAS attB (S1A and S1B Fig), that contained three UAS cassettes for Gal4-mediated transgene expression [14], including one from the pWALIUM vector designed for short hairpin RNA (shRNA) expression [15,16]. Each cassette included a unique multiple cloning site (MCS) flanked by the Gal4 inducible UAS enhancer sequence, followed by the basal promoter of the *hsp70* gene upstream and the SV40 terminator sequence downstream. To simultaneously reduce the activity of multiple tumor suppressor genes, we generated synthetic multi-hairpin shRNA clusters inspired by the microRNA (miRNA) clusters found in most genomes [17,18]. This multigenic vector design allowed targeted expression of up to two oncogenes, each from its own UAS cassette, and simultaneous knockdown of as many as eight genes from a single synthetic multi-hairpin cluster. In other words, a 10-hit cancer model could be generated by a multigenic construct as a single transgenic insertion in the *Drosophila* genome. We have used this approach successfully to generate and drug screen personalized *Drosophila* models built based on tumor genome landscapes of individual patients in the context of a clinical study [14,19].

The multigenic vector design significantly improves our ability to capture the genetic complexity of human disease. Here, we further streamline and expand upon this platform by consolidating the genetic manipulations requiring all three UAS cassettes of pWALIUM 3xUAS attB into a single inducible UAS transgene. This new design combines intron-mediated short hairpin expression [20] and T2A-mediated polycistronic protein expression [21,22] strategies to target multiple genes from a single polycistronic transcript. It frees the other two UAS cassettes of the multigenic vector for additional genetic manipulations to build more sophisticated models from a single vector. We also show that the number of short hairpins targeting each gene of interest can be increased in a single cluster to ensure robust knockdown without limiting the number of genetic manipulations in cancer models; this is particularly useful as validated shRNA sequences with strong *in vivo* efficacy are not available for all genes. Notably, this design can also be used with the standard P-element and phiC31-based vectors for Gal4 regulated transgene expression (pUAST or pUASTattB) and adapted to other targeted expression systems available in *Drosophila* [23–25].

## Results

### Evaluating knockdown efficacy and positional effects within multi[sh] clusters

We have previously demonstrated that eight genes can be effectively knocked down from a single synthetic short-hairpin (sh) cluster [14]. To determine whether knockdown efficacy is altered in the context of a multi[sh] cluster compared to single shRNA expression, we built a series of 4[sh] test clusters by stitching together short hairpin sequences available as transgenic *UAS-shRNA* fly lines from the TRiP collection [26]. To this end, we selected three hairpin sequences targeting the *Drosophila white* (*w*), *singed* (*sn*), and the exogenous *GFP* gene, all of which provided strong knockdown of their target genes at the RNA, protein, or phenotypic level upon ubiquitous expression. We also included a short hairpin targeting the *Drosophila p53* gene, which did not provide significant knockdown as a single hairpin, as a control (Fig 1 and S1A Table). To determine whether there are positional effects within a multi-hairpin cluster that might influence knockdown efficacy, we stitched together these four short hairpins to create four different 4[sh] clusters, *UAS-tester1-4*, where each short hairpin occupied a

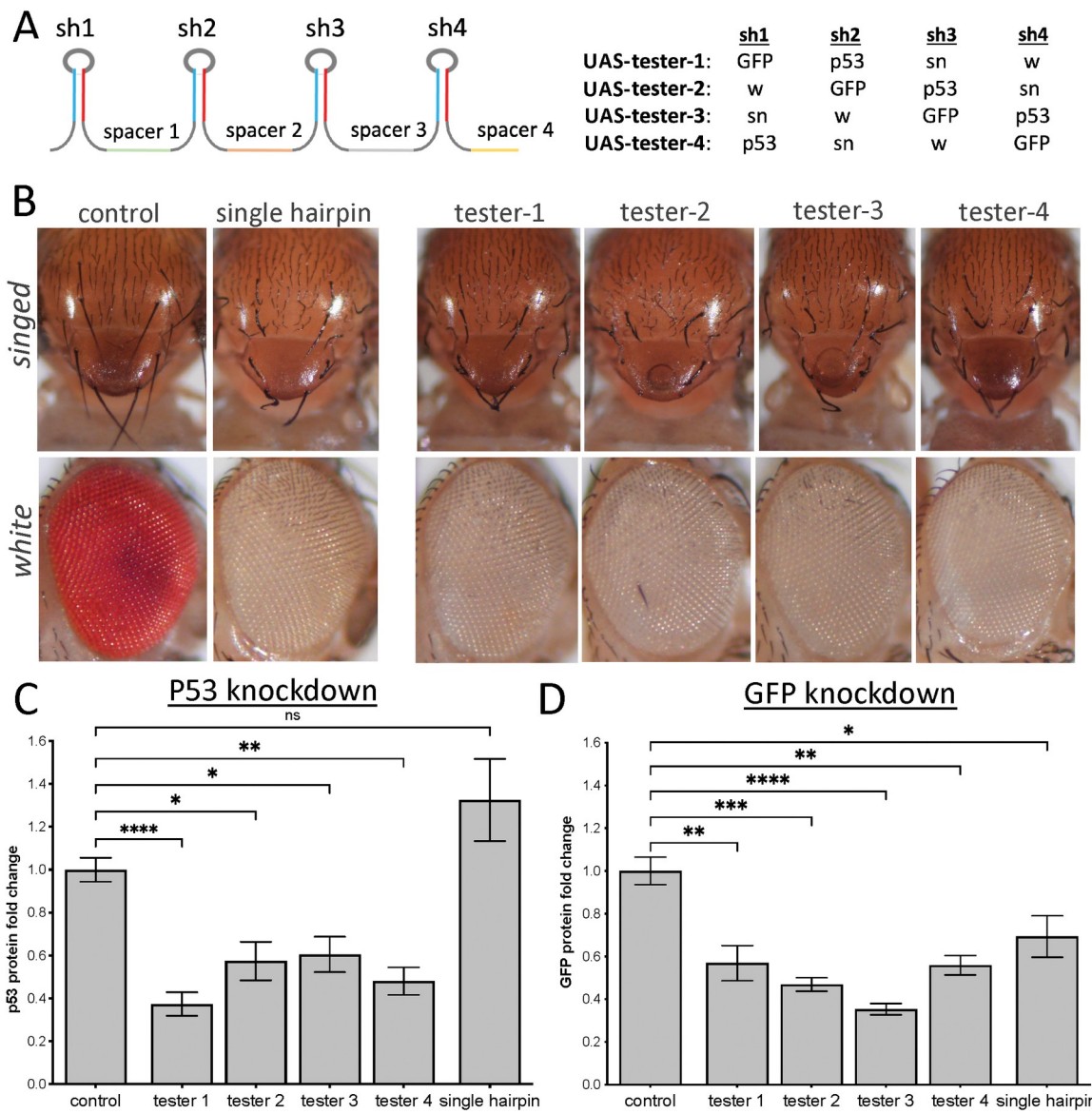

**Fig 1. Evaluating knockdown efficacy and positional effects in the context of synthetic multi-hairpin clusters. A.** 4[sh] tester cluster design. The position of each short hairpin is varied so that all four short hairpins occupy all four possible positions within the cluster. **B.** Evaluation of *sn* and *w* knockdown efficacy using bristle and eye phenotypes observed in response to the ubiquitous expression of tester clusters and single hairpin controls. All phenotypes were fully penetrant (n = 50 animals). *tub-gal4*-only animals were used as baseline controls to demonstrate the wild-type eye and bristle phenotypes. **C,D.** Evaluation of *p53* **(C)** and *GFP* **(D)** knockdown efficacy in response to cluster and single hairpin expression by western blotting. Error bars represent the standard error of the mean (SEM). *:p ≤0.05, **:p≤ 0.01, ***: p≤ 0.001, ****: p≤ 0.0001 (ordinary one-way ANOVA with Tukey's multiple comparisons). Differences between the four tester constructs are not statistically significant.

different position in the cluster (Fig 1A and S1B and S1C Table). We cloned these clusters into our multigenic pWALIUM 3xUAS attB vector [14] (S1A and S1B Fig) and generated transgenic flies. Driving ubiquitous expression of the 4[sh] clusters using *tubulin-gal4* (*tub-gal4*) resulted in a strong knockdown of all targeted genes (Fig 1B–1D and S2A and S2B Table). We did not see a significant difference in knockdown efficiency among the four clusters (Fig 1B–1D), indicating that there are no positional effects, at least in the context of a 4[sh] cluster. Furthermore, cluster-mediated knockdown efficacy was comparable to single hairpin controls,

demonstrating that hairpin expression from a cluster does not reduce its efficacy. Notably, *p53* shRNA expression from all four 4[sh] clusters resulted in strong *p53* knockdown, while the same sequence as a single hairpin did not (Fig 1C). While not generalizable to other hairpins (e.g. Fig 1D), our findings demonstrate that, at least in some cases, short hairpins can be more effective in the context of a synthetic cluster.

To determine whether knockdown efficacy changes in longer synthetic clusters, we generated 8[sh], 12[sh], and 16[sh] clusters by adding additional short hairpins to tester-1 (Fig 2A and S1A and S1D Table), cloned them into pWALIUM 3xUAS attB (S1A and S1B Fig) and established transgenic flies. As simultaneous knockdown of many genes can result in lethality and hinder the evaluation of knockdown efficacy, we prioritized short hairpins targeting exogenous genes frequently used as markers in *Drosophila*, where transgenic UAS constructs to induce their expression for evaluating knockdown efficacy are available. To fill the remaining positions, we selected hairpins targeting *Drosophila* genes whose ubiquitous expression did not result in any phenotype or lethality. Despite these efforts, ubiquitous expression of the longer, 8[sh], 12[sh], and 16[sh] clusters during development resulted in organismal lethality. As a result, we evaluated knockdown efficacy by transiently inducing cluster expression for three days using *tub-gal80^{ts}*, a temperature-sensitive allele of the Gal4 inhibitor Gal80.

We found that increasing cluster length did not affect the efficacy of hairpins present in the original tester-1 cluster, indicating that longer clusters are equally effective in gene knockdown. For instance, both *GFP* and *sn* hairpins, at positions 1 and 3, respectively, provided strong and comparable knockdown in 4[sh], 8[sh], 12[sh] and 16[sh] clusters (Fig 2B and 2C). Hairpins that were only present in longer constructs were also effective: For instance, short hairpins targeting *β-galactosidase (β-gal)* and *sevenless (sev)* genes, present in positions 5 and 7 respectively, resulted in comparable knockdown in response to 8[sh], 12[sh] and 16[sh] cluster expression (Figs 2D and 2E and S2C). We had similar results with hairpins that are present in 12[sh] and 16[sh] (Fig 2F) and in 16[sh] only (Fig 2G). In each case, the expression of a cluster that did not include the tested hairpin was used as a negative control (Fig 2D–2G). Overall, these results demonstrate that cluster length does not negatively impact knockdown efficacy, and at least 16 short hairpins can be expressed from a single synthetic cluster for combinatorial gene knockdowns.

Given the lethality induced by ectopic expression of the longer clusters, we next tested the possibility that the expression of long multi[sh] clusters with many inverted repeats may have some non-specific toxic effects. To this end, we generated two new constructs by inverting the orientation of DNA sequences encoding 8[sh] and 16[sh] clusters to change the transcribed strand and produce a transcript that has the same number and sequence of short hairpins as the original constructs but the opposite orientation of the passenger and guide sequences (S3A Fig and S1 Table). We cloned the inverted clusters into pWALIUM 3xUAS attB and generated transgenic flies. Ubiquitous expression of these inverted clusters had no effect on viability (S3B Fig), suggesting that the lethality observed with the original constructs is likely a synergistic response to the simultaneous knockdown of multiple genes rather than general toxicity of long multi[sh] cluster expression. Furthermore, an 8[sh] cluster targeting the *p53* gene, which is not required for development [27], did not result in any lethality despite providing strong *p53* knock-down at the protein and RNA level (S3C–S3F Fig), further supporting the conclusion that long hairpin clusters are not inherently toxic to the cell.

These findings demonstrate that while feasible, simultaneous and ubiquitous knock-down of a large number of genes could result in synthetic lethality, potentially limiting the use of longer clusters in biological studies. However, long clusters have additional, more broadly relevant applications, including as an *in vivo* hairpin discovery tool to identify effective hairpins for a large number of genes for subsequent studies, calibrating the level of knockdown of a gene by varying the number of hairpins, or targeting fewer genes with multiple short hairpins each, as

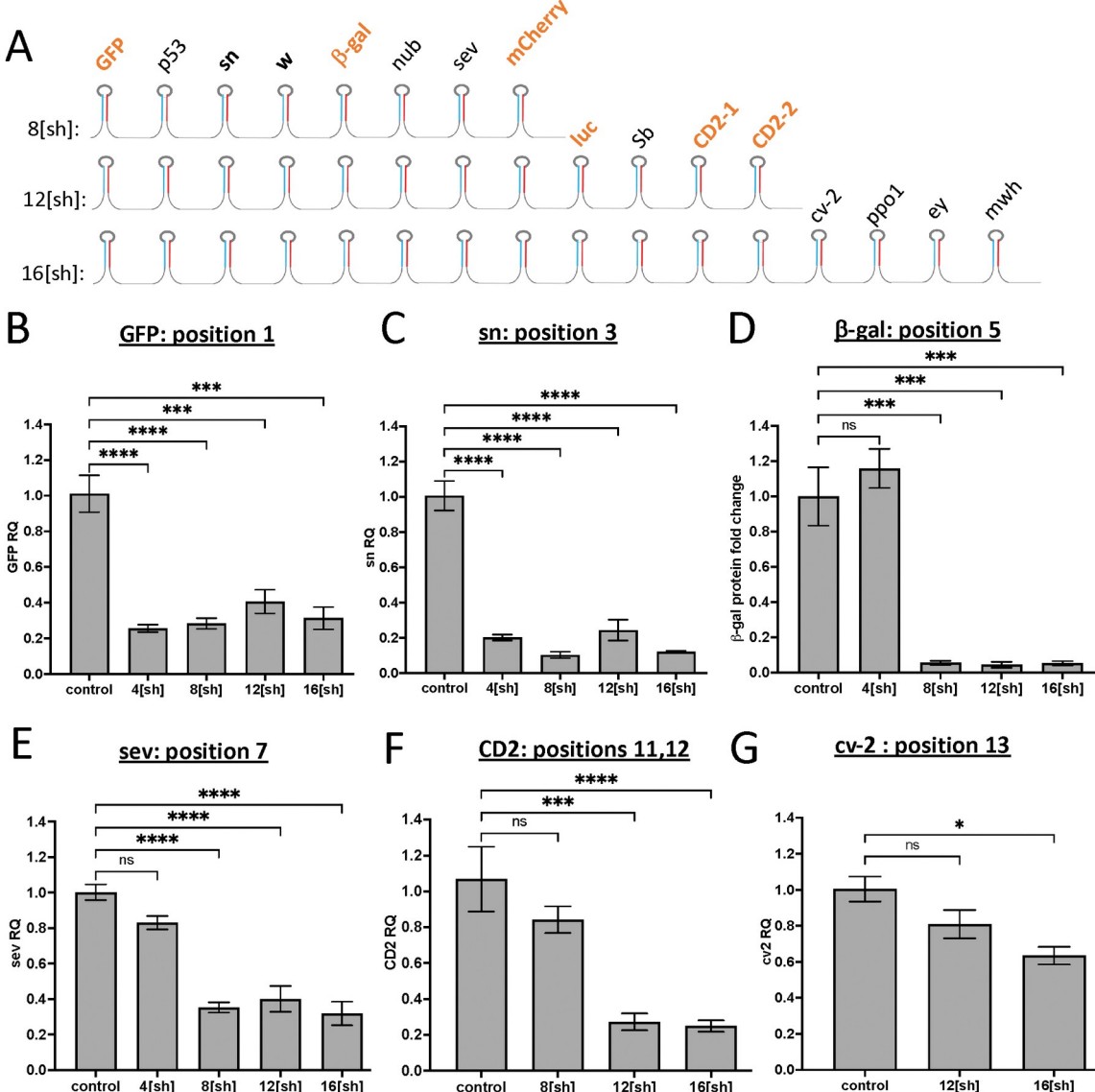

**Fig 2. Evaluating knockdown efficacy in the context of longer clusters. A**. Longer short hairpin cluster designs. 8[sh], 12[sh], 16[sh] clusters were generated by adding additional hairpins to the 4[sh] tester-1 cluster. Exogenous genes targeted by the clusters are in orange. **B-G.** Knockdown efficacy of short hairpins at indicated positions within longer clusters by qPCR **(B,C,E-G)** and western blot **(D)** analysis. Clusters that do not include the hairpin being evaluated are used as negative controls (4[sh] for **D** and **E**, 8[sh] for **F**, 12 [sh] for **G**). Error bars represent the standard error of the mean (SEM). *:p ≤0.05, **:p≤ 0.01, ***: p≤ 0.001, ****: p≤ 0.0001 (ordinary one-way ANOVA with Tukey's multiple comparisons).

an *in vivo*, transgenic equivalent of siRNA pools commonly used in cell-based assays. We next explored some of these applications focusing on *Drosophila* orthologs of genes commonly mutated in colorectal cancer.

## Using the multi[sh] design to screen hairpins targeting colon cancer tumor suppressors

Sequencing and genome-wide association studies often identify genes whose orthologs are not well-characterized and for which functionally validated short hairpins are not available. Multi

[sh] clusters offer a streamlined way to screen short hairpins targeting several genes for *in vivo* efficacy without the need for generating separate transgenic animals for each. To test our approach using disease-relevant genes, we built a series of multi[sh] constructs targeting *Drosophila* orthologs of four tumor suppressors recurrently mutated in colon cancer: *TP53*, *APC*, *SMAD4* and *SMAD2* (*Drosophila* orthologs *p53*, *apc*, *Med* and *smox*, respectively). First, we generated four different 4[sh] clusters where each gene is targeted with a different hairpin sequence, including one, pams-4, constructed using hairpins that were predicted to be moderately effective as a control (Fig 3A and S2A Table). Hairpins were selected using the publicly available shRNA prediction tool DSIR [28,29] as previously described [15]. We cloned each cluster into pWALIUM 3xUAS attB (S1A and S1B Fig) and generated transgenic flies. We then tested the knockdown efficiency of each short hairpin *in vivo* by ubiquitous transient expression of the clusters. Using 4 transgenic fly lines, this approach allowed *in vivo* evaluation of 16 different short hairpins to identify multiple short hairpins that provided a significant knockdown for each gene (Fig 3B). As predicted, the pams-4 control was the least effective cluster.

Next, we tested whether we could calibrate the level of knockdown by varying the number of short hairpins targeting a gene. To this end, we stitched pams-2 and pams-3 clusters (Fig 3A) together by PCR to generate the pams[8sh$^{2-3}$] construct that includes two different short hairpins for each targeted gene (Fig 3C). We cloned the new cluster into pWALIUM 3xUAS attB (S1A and S1B Fig) and generated transgenic flies. We then compared the level of knockdown induced by the pams[8sh$^{2-3}$] and the two 4[sh] clusters pams-2 and pams-3. Bringing together two hairpins targeting each gene did not increase the level of knockdown (Fig 3D), suggesting that there are no additive effects between multiple hairpins targeting the same gene, at least in the case of these four genes we analyzed. Regardless, these results demonstrate that the multi[sh] clusters are useful to evaluate the efficacy of multiple hairpins *in vivo* and identify optimal hairpin sequences to build new clusters for disease modeling or functional studies.

Even if including multiple hairpins targeting a single gene does not result in a progressively stronger knockdown, it may increase the likelihood that at least one hairpin with strong i*n vivo* efficacy will be present in the final cluster, eliminating the need to create new clusters using hairpins identified from the initial screen. To further explore this possibility, we created a new 16[sh] cluster targeting the same 4 genes with four different short hairpin sequences for each gene, using a combination of short hairpins from previous pams constructs and new hairpins designed using DSIR (Fig 3A and 3E and S2B Table). We cloned the pams16[sh] cluster into pWALIUM 3xUAS attB (S1A and S1B Fig) and generated transgenic flies. The pams16[sh] construct provided a strong knockdown of all four genes (Fig 3F), demonstrating that longer clusters can be beneficial for targeting a small number of genes, as multiple hairpins targeting each gene of interest can be included in the design. Overall, these findings demonstrate the utility of this approach in evaluating the efficacy of a large number of hairpins *in vivo* to optimize knockdown efficacy and as a genetically encoded pooled shRNA strategy to increase the likelihood of strong knockdown.

## A polycistronic design for combinatorial gene expression and knockdown

Our multigenic vector design allows multi-gene expression and knockdown from a plasmid with three separate UAS cassettes (S1A Fig). We next sought to consolidate these into a polycistronic transgene that expresses a multi[sh] cluster from within an intron placed upstream of coding sequences of multiple proteins separated by a T2A sequence from a single UAS cassette. Intron-mediated expression of single short hairpins has been documented before [20], and T2A-mediated expression of multiple proteins is a relatively common strategy in *Drosophila*

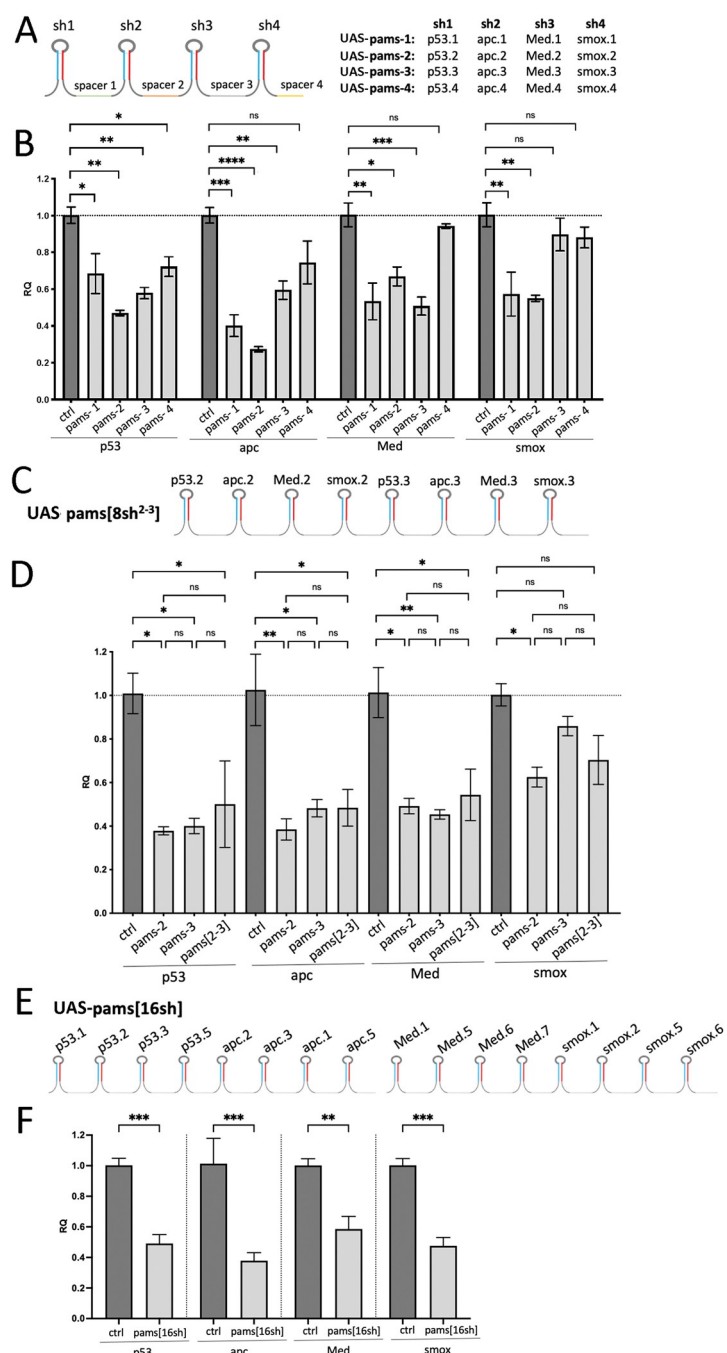

**Fig 3. Multi[sh] cluster design as a streamlined screening strategy to identify effective short hairpins for multiple tumor suppressors. A**. Four different 4[sh] clusters targeting *Drosophila* orthologs of four recurrently mutated colorectal cancer tumor suppressors *TP53* (*p53*), *APC* (*apc*), *SMAD4* (*Med*) and *SMAD2* (*smox*). Each cluster expresses a different short hairpin targeting the same set of four genes. **B**. Evaluation of knockdown efficacy of *p53*, *apc*, *Med* and *smox* in response to the ubiquitous expression of the 4[sh] pams clusters by qPCR. **C**. An 8[sh] cluster generated by stitching pams-2 and pams-3 4[sh] clusters. **D**. Evaluation of knockdown efficacy of *p53*, *apc*, *Med* and s*mox* in response to the ubiquitous expression of the 8[sh] pams cluster compared to pams-2 and pams-3 by qPCR. **E**. A 16[sh] cluster which includes 4 different hairpin sequences targeting each gene. **F**. Evaluation of knockdown efficacy of *p53*, *apc*, *Med* and *smox* in response to the ubiquitous expression of the 16[sh] pams cluster by qPCR. **B,D,F.** Error bars represent the standard error of the mean (SEM). *:p ≤0.05, **:p≤ 0.01, ***: p≤ 0.001, ****: p≤ 0.0001 (ordinary one-way ANOVA with Tukey's multiple comparisons).

[21,22]. To combine these two elements with our multi[sh] cluster strategy, we first designed a new UAS-cassette with two distinct multiple cloning sites, one within the *Drosophila ftz* intron to allow intron-mediated shRNA expression [20], and a second one to introduce protein-coding sequences downstream of the intron (S1C Fig). We then generated a new multigenic vector, pWALIUM-intron 3xUAS attB, that includes this polycistronic UAS-cassette along with two standard ones to allow expression of additional transgenes (S1D Fig).

To determine whether intron-mediated multi[sh] cluster expression altered knockdown efficacy, we cloned the tester-1 cluster (Fig 1A) into the cluster cloning site within the *ftz* intron and the *mCherry* coding sequence into the CDS cloning site downstream of the intron (Fig 4A) in pWALIUM-intron 3xUAS attB (S1C and S1D Fig) and generated transgenic flies. Ubiquitous expression of this transgene resulted in strong gene knockdown comparable to non-intron mediated expression of the same cluster from the original pWALIUM 3xUAS attB vector (Fig 4B), demonstrating that the *ftz* intron can be used as a structural design element to facilitate co-expression of a hairpin cluster and a protein as a single transcript.

To test the ectopic expression of multiple proteins, we selected two oncogenes commonly altered in human colon tumors, $dRAS^{G12V}$, which represents a commonly observed oncogenic mutation in *KRAS*, and *chico*, the *Drosophila* ortholog of *IRS2*, a gene that frequently shows copy number gain in human colon tumors [30,31]. $dRAS^{G12V}$ and *chico* coding sequences were tagged by FLAG and HA epitopes, respectively, to facilitate detection by western blot analysis. We created a single bicistronic transgene where the two coding sequences, separated by the T2A sequence (*chico—HA—T2A - FLAG—dRAS$^{G12V}$*), were tandemly cloned into MCS1 of pWALIUM 3xUAS attB (Figs 4C and S1A). As a control, we separately cloned each tagged coding sequence downstream of its own UAS promoter into MCS1 and MCS2 of pWALIUM 3xUAS attB (*UAS-chico–HA; UAS-FLAG—dRAS$^{G12V}$*, Figs 4C and S1A) and generated transgenic flies. Transient ubiquitous expression of the bicistronic and control transgenes resulted in strong expression of both proteins (Fig 4D).

Next, we tested the final polycistronic transgene design by combining the short hairpin cluster and T2A-mediated bicistronic protein expression into a single UAS-inducible transgene that represents a 6-hit colorectal cancer model expressing 2 oncogenes and knockdown of 4 tumor suppressors. To this end, we cloned the 4[sh] cluster pams-1 and the [*chico—HA—T2A - FLAG—dRAS$^{G12V}$*] fragment into the polycistronic UAS cassette of our new pWALIUM-intron 3xUAS attB vector (Fig 4E and S1C and S1D Fig) and generated transgenic flies. Transient ubiquitous expression of this polycistronic transgene resulted in significant knockdown of *p53*, *apc*, *Med* and *smox* (Fig 4F) and ectopic expression of both Chico and dRAS$^{G12V}$ proteins (Fig 4G). The use of a single transgene to manipulate 6 different genes from a single polycistronic transcript frees the remaining two cassettes of the multigenic vector for additional transgenes. These cassettes can be used to clone additional polycistronic transgenes to further increase the complexity of disease models or incorporate some of the components required for inducible genetic tools to further reduce the number of independent transgenic insertions required in experimental animals. Lastly, the polycistronic transgene design is compatible with the standard single-cassette vectors typically used for transgenesis in *Drosophila* and can be adapted for other experimental systems to perform combinatorial genetic manipulations without the need for a multigenic vector.

## Discussion

In recent years, *in vivo* combinatorial genetic perturbations have become increasingly essential to generate experimental models that capture the genetic complexity and heterogeneity of human diseases and for detailed mechanistic explorations of local and systemic interactions

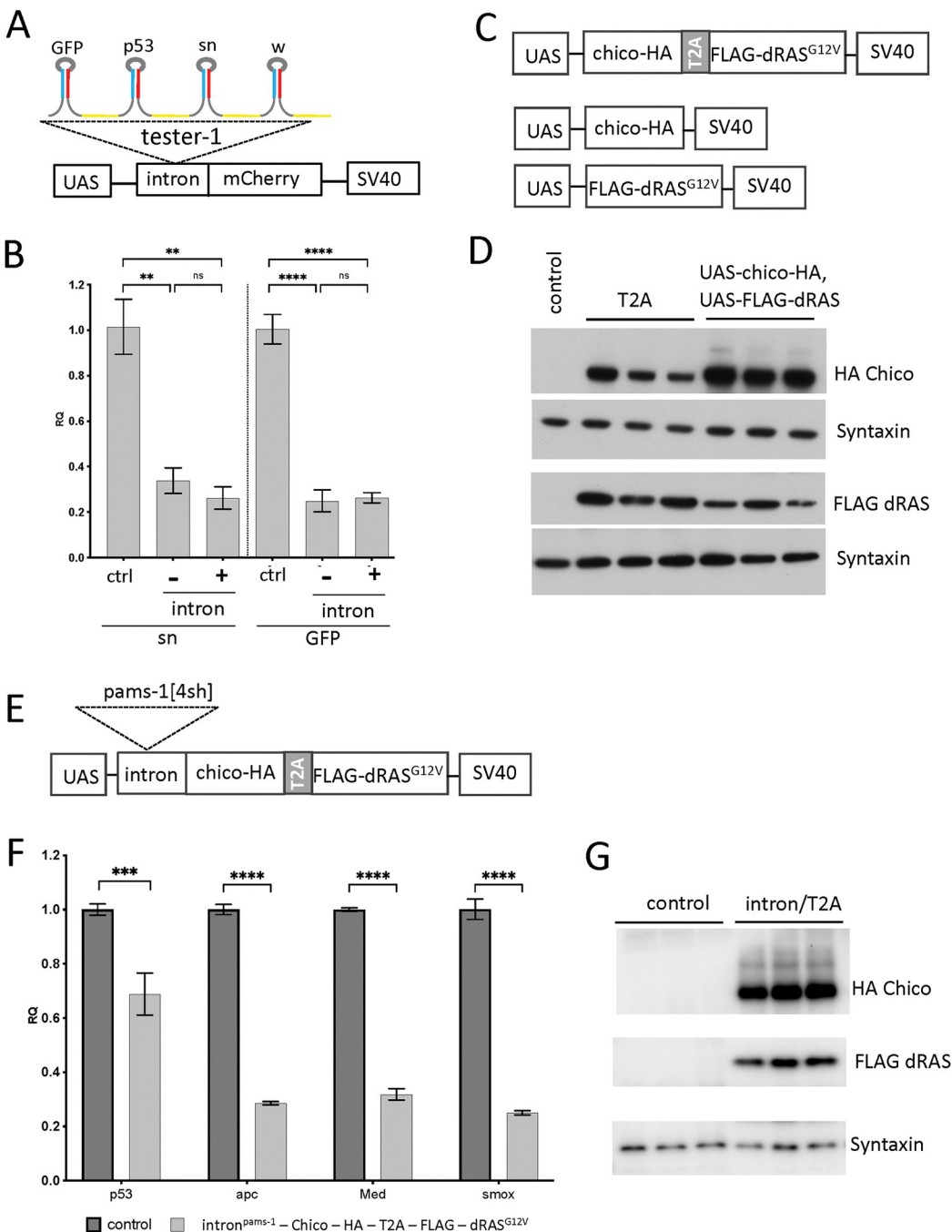

**Fig 4. A polycistronic design for simultaneous gene knockdown and expression. A**. The construct designed for intron-mediated expression of the 4[sh] tester 1 cluster. **B.** Evaluation of *sn* and *GFP* knockdown in response to intron-mediated cluster expression. **C.** The bicistronic construct designed for T2A-mediated expression of *Drosophila* orthologs of two oncogenes commonly altered in human colon tumors, an oncogenic *KRAS[G12V]* (*dRAS^{G12V}*) and *IRS2* (*chico*). The control construct where each coding sequence is cloned downstream of its own UAS enhancer/promoter in the multigenic vector. **D.** Evaluation of dRAS^{G12V} and Chico protein expression from the bicistronic construct by western blot analysis. **E.** The polycistronic construct designed for knockdowns of the 4 colorectal cancer tumor suppressors by intron-mediated expression of a 4[sh] cluster and T2A-mediated expression of two proteins from a single transcript. **F,G.** Evaluation of knockdown efficacy of *p53*, *apc*, *Med* and *smox* by qPCR (**F**) and dRAS^{G12V} and Chico protein expression (**G**) upon ubiquitous expression of the 6-hit polycistronic construct (**E**). **B,F.** Error bars represent the standard error of the mean (SEM). **:$p \leq 0.01$, ***: $p \leq 0.001$, ****: $p \leq 0.0001$ (ordinary one-way ANOVA with Tukey's multiple comparisons).

underlying disease states. Here, we report a new design that combines elements from different genetic manipulation tools for targeted, inducible expression and knockdown of multiple genes. This highly flexible and adaptable tool consolidates multiple genetic perturbations into a single, inducible polycistronic transgene to further increase the genetic sophistication of disease models. The ability to use a single UAS cassette for multiple genetic manipulations provides an opportunity to introduce additional transgenes, reporters, and lineage-tracing constructs into the multigenic vector. It also makes it broadly adaptable to other targeted expression, site-specific recombination, and gene editing platforms for mechanistic studies.

Multi-omics approaches to studying disease states often identify complex gene signatures as reporters of disease progression, recurrence, drug response, and resistance [32,33]. Exploring the functional relevance of multigenic signatures and their potential contributions to disease in experimental models that reflect the genetic complexity and heterogeneity of human disease can be challenging. Using this design, combinatorial genetic perturbations reflecting such multigenic signatures can be more easily generated to investigate their roles in disease development or drug response.

The rapid advancement of sequencing technologies and the concomitant reduction of cost led to the identification of a large number of variants in human genes associated with human disease [34]. However, functional exploration of these variants to investigate causal relationships in disease development and to explore underlying molecular mechanisms has been difficult. This is a particular challenge in cancer: tumor sequencing studies routinely identify large numbers of novel variants in potentially cancer-relevant genes whose functional relevance can be challenging to investigate [35–37] as they require representative disease models in genetically tractable experimental systems. "Humanization" of fly genes has been a successful approach for functional studies of disease-associated human genes and variants, where a disease-associated human gene or variant is expressed while simultaneously removing its *Drosophila* ortholog [38–40]. The polycistronic design described here offers another useful strategy for the humanization of *Drosophila* genes by inducible expression of a single transcript that expresses the human variant while simultaneously knocking down its *Drosophila* ortholog. It can be particularly beneficial in cases where multiple *Drosophila* orthologs with redundant functions must be simultaneously removed to study the function of a human gene. It could also be used to express multiple human genes and reduce the activity of their *Drosophila* orthologs to "humanize" disease-relevant regulatory nodes and investigate their functions.

Recent advances in CRISPR-based technologies have made somatic, inducible gene editing possible in multiple model systems, and multiplexed DNA editing technologies that allow simultaneous editing of multiple genes are promising for modeling polygenic diseases [41]. However, significant challenges associated with editing multiple genes remain. These include reduced efficiency of editing due to competition between individual guide RNAs [42–44], undesired chromosomal rearrangements as a result of cuts at multiple genomic locations [45–47], and potential activation of DNA damage response pathways [48,49] that can reduce the fitness of targeted cells and modify disease phenotypes. Importantly, as with CRISPR-based multi-gene editing, multiplexed shRNA approaches for gene knockdown can result in enhanced off-target effects [46,47]; therefore, regardless of the approach, carefully controlled experiments to validate these tools and rule out potential off-target effects are essential. Notably, our polycistronic design allows us to design rescue constructs that co-express one or more genes targeted by the hairpin clusters for validation purposes. As the research community attempts to address and overcome technical and biological challenges associated with multiplexed gene-editing technologies, shRNA-mediated knockdown and targeted gene expression strategies in this design offer a well-established complementary approach to combinatorial gene editing, which can be adapted for use in other experimental systems.

## Materials and methods

### Cloning and Transgenesis

The 4[sh] clusters tester1-4 were synthesized by four rounds of PCRs using the primers listed in S1C Table. Guide and spacer sequences used in the construction are shown in S1A Table. Digitally assembled clusters and individual hairpins are shown in S1B Table. PCR products representing fully assembled [4sh] clusters were digested with XbaI (5') and EcoRI (3') introduced during the final round of PCR and cloned into the pWALIUM-Multiple Cloning Site (MCS) of pWALIUM 3xUAS attB, which is designed for shRNA expression (S1A and S1B Fig). To generate the longer test clusters 8[sh], 12[sh], and 16[sh], three additional 4[sh] clusters representing hairpins 5–8, 9–12, and 13–16 were digitally assembled (S1D Table) and generated by gene synthesis (Genewiz). The sequence-confirmed fragments were PCR-amplified and stitched to the 4[sh] cluster tester-1 by sequential PCR reactions. The resulting 8[sh], 12 [sh], and 16[sh] clusters were digested with XbaI (5') and NotI (3') and cloned into the pWAL-MCS of pWALIUM 3xUAS attB (S1A and S1B Fig). Inverted constructs were generated by PCR-amplifying the 8[sh] and 16[sh] clusters from the multigenic vectors used for transgenesis using a new set of primers designed to append NotI and EcoRI restriction enzyme sites at the 5' and the 3' ends of the clusters respectively so that the clusters could be re-cloned into the MCS of the multigenic vector in the inverted orientation. The final products were sequence confirmed (Genewiz), and transgenic flies were generated by PhiC31-mediated targeted integration into the *attp2* landing site on the third *Drosophila* chromosome [13] (BestGene).

Synthesis of 4[sh] pams1-4 clusters (S2A Table) and the 4[sh] clusters targeting *p53*, *apc*, *Med*, and *smox* that make up the pams [16sh] cluster (S2B Table) were outsourced to Genewiz. Sequence confirmed 4[sh] pams clusters were cloned into the multigenic vector using XbaI (5') and EcoRI (3'). 4[sh] *p53*, *apc*, *Med*, and *smox* clusters were stitched together by sequential PCRs to create the final [16sh] pams cluster, which was then cloned into the pWAL-MCS of pWALIUM 3xUAS attB using XbaI (5') and NotI (3') (S1A and S1B Fig). pams-2 and pams-3 4[sh] clusters were similarly stitched together to create the pams[8sh$^{2-3}$] cluster, which was cloned into the pWAL-MCS of pWALIUM 3xUAS attB using XbaI (5') and EcoRI (3') (S1A and S1B Fig). All constructs were sequence verified before transgenesis. Transgenic flies were generated by PhiC31-mediated targeted integration [13]. Vectors carrying the 4[sh] pams1-4 and the pams[8sh$^{2-3}$] clusters were inserted into the attp40 landing site on the 2nd chromosome and the pams [16sh] vector into the attp2 landing site on the third chromosome (Bestgene).

Validation constructs for T2A mediated protein expression were generated as follows: First, a fragment that contained the T2A sequence flanked by the 3xHA tag sequence at the 5' end and the 3xFLAG tag sequence at the 3' end was digitally assembled using sequences from Addgene plasmids #24355, #47948 and #32426 for 3xHA, 3xFLAG, and T2A respectively and generated by gene synthesis (Genewiz). The sequence-confirmed final product [AgeI - 3xHA—T2A - 3xFLAG], and the *dRAS*$^{G12V}$ coding sequence from a vector available in-house were PCR-amplified separately and stitched together in another PCR reaction. The stitched product, [AgeI - 3xHA—T2A - 3xFLAG—dRAS$^{G12V}$ - PacI], was cloned into the MCS1 of pWALIUM 3xUAS attB using AgeI (5') and PacI (3') (S1A Fig) and sequence verified. Next, the *chico* coding sequence was amplified from a vector available in-house using primers that appended a FseI site to the 5' end, removed the stop codon, and appended an AgeI site to the 3' end. The resulting product was cloned into the multigenic vector upstream of [AgeI - 3xHA—T2A - 3xFLAG—dRAS$^{G12V}$ - PacI] using FseI (5') and AgeI (3') to create the final [UAS-chico-3xHA—T2A - 3xFLAG—dRAS$^{G12V}$ - SV40] construct. The sequence-verified final construct

was also used to PCR amplify individual tagged coding sequences [FseI—chico- 3xHA—PacI] and [BsiWI 3xFLAG—dRAS^G12V - PmeI] for cloning into the MCS1 and MCS2 of pWALIUM 3xUAS attB respectively (S1A Fig) to create [UAS—FseI—chico- 3xHA—PacI—SV40; UAS—BsiWI 3xFLAG—dRAS^G12V - PmeI—SV40]. Transgenic flies carrying each construct were generated by PhiC31-mediated targeted integration into the *attp2* landing site on chromosome three [13] (BestGene).

## pWALIUM-intron 3xUAS attB vector construction

For intron-mediated cluster and protein co-expression, we designed a new UAS cassette, [10XUAS—hsp70 basal promoter—ftz intron(cluster MCS: Acc65I, KpnI, XhoI)–CDS MCS: NotI, XbaI], which included 1) a longer *ftz* intron sequence previously used for intron-mediated short hairpin expression with an MCS within the intron to clone short hairpin sequences [20], and 2) another MCS for cloning a protein-coding sequence (CDS) immediately downstream of the intron (S1C Fig). The new cassette was generated by gene synthesis (Genewiz) and used to create a new multigenic vector, pWALIUM-intron 3xUAS attB, along with two additional UAS cassettes with their own unique MCS for cloning additional transgenes (S1C and S1D Fig).

To test intron-mediated cluster expression, *mCherry* coding sequence was PCR-amplified from a transgenic *UAS-mCherry* line available in-house using primers designed to append NotI and XbaI restriction sites to the 5' and 3'ends of the fragment, respectively, and cloned into the CDS cloning site downstream of the *ftz* intron (S1C Fig). The tester-1 4[sh] cluster was PCR-amplified using a new set of primers designed to append KpnI and XhoI to the 5' and 3' ends, respectively, and cloned into the cluster cloning site within the intron (S1C Fig). The final product vector was confirmed by sequencing. Transgenic flies were generated by targeted insertion into the attp2 landing site on the 3rd chromosome (Bestgene).

To generate the polycistronic construct that combines colon cancer tumor suppressor knockdown and oncogene expression, the pams-1 [4sh] cluster was PCR-amplified from pWALIUM 3xUAS attB with a new set of primers to append Kpn I and XhoI restriction sites at the 5' and 3' ends of the cluster respectively and cloned into the cluster cloning MCS within the ftz intron of pWALIUM-intron 3xUAS attB using Kpn I(5') and XhoI (3') (S1C Fig). The [chico- 3xHA—T2A - 3xFLAG—dRAS^G12V] fragment was PCR amplified from pWALIUM 3xUAS attB using primers designed to append NotI and XbaI sites to the 5' and 3' ends of the fragment and cloned into the CDS cloning site of pWALIUM-intron 3xUAS attB downstream of the *ftz* intron using those enzymes (S1C Fig). The final sequence verified vector was used to generate transgenic flies by PhiC31-mediated targeted integration into the attp2 landing site on chromosome three [13] (BestGene).

## *Drosophila* crosses and phenotypic analyses

Experimental animals used in this study were generated by crossing male flies carrying the UAS-transgenes described in the previous section to virgin females with the genotype *w^1118*; *tub-gal4 tub-gal80^ts /TM6b, Hu, Tb. tub-gal4* drives UAS-transgene expression ubiquitously and *tub-gal80^ts*[7] was used temporally regulate Gal4 activity. All crosses were set up on Bloomington's semi-defined medium. Crosses to generate adult experimental animals for phenotypic and survival analyses were set up at 29°C to induce transgene expression throughout development. Experimental animals for qPCR and western blot analyses were generated from crosses set up at 18°C to keep transgenes silent during early development and shifted to 29°C to transiently induce transgene expression for 3 days during larval development.

## qPCR and western blot analysis

RNA and protein were extracted from whole experimental and control larvae at the late 3rd instar stage using previously established protocols [14,50] (6 larvae/biological replicate; 3 biological replicates/genotype). For evaluating the knockdown efficacy of exogenous proteins, animals ubiquitously expressing GFP (Fig 1D), β-gal (Fig 2D), and CD2 (Fig 3F) were used as controls. For all others, $w^{1118}$; *tub-gal4 tub-gal80$^{ts}$* /+ animals were used as controls. Western blot analyses were performed as described previously [14,50] using the following primary and secondary antibodies: mouse anti-OctA(FLAG) antibody (1:1000, Santa Cruz Biotechnology, sc-166355), rat anti-HA High-Affinity Antibody (1:1000, ROAHAHA, MilliporeSigma), rabbit anti-GFP N-terminal antibody (1:4000, G1544, MilliporeSigma), mouse anti-p53 antibody (1:1000, Developmental Studies Hybridoma Bank, #dmp53 H3), and mouse anti-Syntaxin antibody (1:1000, Developmental Studies Hybridoma Bank, #8C3) as a loading control, goat anti-mouse and goat-anti-rat IgG HRP-linked secondary antibodies (1:5000, Cell Signaling Technology #7076 and #7077, respectively). qPCR analyses were performed as described previously [14] using the housekeeping gene *rpl32* as an internal control. qPCR data were analyzed using the double-ΔCT method [51].

## Supporting information

**S1 Fig. Multigenic vectors used in the study. A.** Map of our previously published pWALIUM 3xUAS attB vector, which includes three UAS cassettes, each with its unique multiple cloning site (MCS). **B.** Detailed map of the pWALIUM-derived cassette of pWALIUM 3xUAS attB, which was used to clone the test clusters in this study. **C.** The new polycistronic UAS cassette designed for intron-mediated expression of a short-hairpin cluster and a protein-coding sequence as a single transcript generated in this study. **D.** Map of the new pWALIUM-intron 3xUAS attB multigenic vector which includes the new cassette (**C**) along with two other standard cassettes (UAS-MCS1 and UAS-MCS2). Complete sequences of both plasmids are provided in S1 Data and S2 Data.
(TIFF)

**S2 Fig. Western blots used for quantification in Figs 1 and 2.** Western blot images used for generating the data presented in Fig 1C (**A**), 1D (**B**), and 2D (**C**). Syntaxin (Syt) was used as loading control.
(TIFF)

**S3 Fig. Evaluating long cluster toxicity using control clusters. A**. The inverted cluster design is illustrated using the 8[sh]-INV construct as an example. The inverted constructs are designed to generate the same number and sequence of hairpins when expressed but do not target any genes. **B**. Ubiquitous expression of the inverted 8[sh] and 16[sh] clusters does not result in organismal lethality. **C-F.** An 8-hairpin cluster targeting p53 with strong efficacy at the RNA (**C**), and protein level (**D,E**) does not result in organismal lethality upon ubiquitous expression during development (**F**).
(TIFF)

**S1 Table. Tester construct designs and sequences. A**. Guide sequences and their sources for each gene targeted in test 4[sh], 8[sh], 12[sh], 16[sh] clusters (top) and 39 nucleotide-long sequences selected as spacers to separate individual hairpins (bottom). Spacer sequences are derived from well-characterized *Drosophila* microRNA genes as indicated. **B.** 4[sh] test cluster design. Sequences that make up each short hairpin (5' flank, passenger, loop, guide and 3' flank), concatenated short-hairpin sequences, and the full clusters, including the spacer

sequences between individual short hairpins are provided. **C.** Primers for 4 sequential PCR reactions used to synthesize the 4[sh] test clusters. **D.** Additional 4[sh] clusters designed to append to tester-1 to generate 8[sh], 12[sh] and 16[sh] test clusters.
(XLSX)

**S2 Table. Proof of concept pams construct designs and sequences. A.** Hairpin and cluster sequences of the four 4[sh] pams constructs pams1-4. **B.** Hairpin and cluster sequences of the 16[sh] pams constructs. Spacer sequences that separate individual hairpins can be found in S1A Table.
(XLSX)

**S1 Data. Complete sequence and map of the pWALIUM 3xUAS attB vector.**
(DNA)

**S2 Data. Complete sequence and map of the pWALIUM-intron 3xUAS attB vector.**
(DNA)

**S3 Data. Numerical data underlying graphs and summary statistics presented in the main figures.**
(XLSX)

# Acknowledgments

We thank Dr. Brian Washburn, Dr. Diego Zorio, and Cheryl Pye at the Florida State University Molecular Cloning Facility and Jason Cassara for technical support. This study used transgenic RNAi lines (Office of the Director R24 OD030002: "TRiP resources for modeling human disease") obtained from the Bloomington *Drosophila* Stock Center (NIH P40OD018537).

# Author Contributions

**Conceptualization:** Ross L. Cagan, Erdem Bangi.

**Formal analysis:** Erdem Bangi.

**Funding acquisition:** Erdem Bangi.

**Investigation:** Alexander G. Teague, Maria Quintero, Fateme Karimi Dermani, Erdem Bangi.

**Methodology:** Alexander G. Teague, Erdem Bangi.

**Project administration:** Erdem Bangi.

**Writing – original draft:** Erdem Bangi.

**Writing – review & editing:** Maria Quintero, Fateme Karimi Dermani, Ross L. Cagan, Erdem Bangi.

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
