## [Decision Letter · Decision Letter 0]

18 May 2023

Dear Dr Bangi,

We are pleased to inform you that your manuscript entitled "A polycistronic transgene design for combinatorial genetic perturbations from a single transcript in Drosophila" has been editorially accepted for publication in PLOS Genetics. Congratulations!

Yours sincerely,

Ken M. Cadigan

Academic Editor

PLOS Genetics

Quanjiang Ji

Section Editor

PLOS Genetics

Comments from the reviewers (if applicable):

Reviewer's Responses to Questions

**Comments to the Authors:**

Reviewer #1: I had a look at the revised manuscript from Dr. Bangi and colleagues. The authors addressed my previous concern regarding the multigenic vector(s) appropriately by providing a figure, annotated maps and using clear terminology. They also present the issue of longer hairpin clusters in a more balanced way and mention the potential limitations of longer arrays. I believe that the presented tools (laborious to generate) are of sufficient novelty and well suited to study more complex disease models or developmental processes. I reconfirm - in line with the initial conclusion drawn by reviewer three - that the improved manuscript by Teague et al. warrants publication in PLoS Genetics.

Reviewer #2: In this revised version of the article, Teague et al. address most of the issues that I and the other reviewers raised previously. In particular, they address my main concern that the long hairpin clusters may have a non-specific deleterious effect on animal viability. As before the manuscript is well-written and the results clearly presented.

However, I remain unconvinced that the new vector represents a substantial advance over previous methods, and that it is likely to be adopted by many researchers.

Reviewer #3: The authors addressed all of my concerns and comments.

**Have all data underlying the figures and results presented in the manuscript been provided?**

Reviewer #1: None

Reviewer #2: Yes

Reviewer #3: Yes

PLOS authors have the option to publish the peer review history of their article (what does this mean?). If published, this will include your full peer review and any attached files.

Reviewer #1: No

Reviewer #2: No

Reviewer #3: No

**Data Deposition**

http://datadryad.org/submit?journalID=pgenetics&manu=PGENETICS-D-23-00397

**Press Queries**

---

## [Editor Report · Acceptance letter]

31 May 2023

PGENETICS-D-23-00397 

A polycistronic transgene design for combinatorial genetic perturbations from a single transcript in Drosophila 

Dear Dr Bangi, 

We are pleased to inform you that your manuscript entitled "A polycistronic transgene design for combinatorial genetic perturbations from a single transcript in Drosophila" has been formally accepted for publication in PLOS Genetics! Your manuscript is now with our production department and you will be notified of the publication date in due course.

With kind regards,

Zsofia Freund

PLOS Genetics

On behalf of:
